# The Dilemma of Falls in Older Persons: Never Forget to Investigate the Syncope

**DOI:** 10.3390/medicina57060623

**Published:** 2021-06-15

**Authors:** Francesca Perego, Beatrice De Maria, Laura Bagnara, Valeria De Grazia, Mauro Monelli, Matteo Cesari, Laura Adelaide Dalla Vecchia

**Affiliations:** 1IRCCS Istituti Clinici Scientifici Maugeri, 20138 Milan, Italy; francesca.perego@icsmaugeri.it (F.P.); beatrice.demaria@icsmaugeri.it (B.D.M.); laura.bagnara@icsmaugeri.it (L.B.); valeria.degrazia@icsmaugeri.it (V.D.G.); mauro.monelli@icsmaugeri.it (M.M.); matteo.cesari@unimi.it (M.C.); 2Department of Clinical Sciences and Community Health, Università Degli Studi di Milano, 20122 Milan, Italy

**Keywords:** syncope, fall, older persons, orthostatic hypotension, disability, emergency department

## Abstract

*Background and objectives*: Falls represent a major cause of morbidity, hospitalizations, and mortality in older persons. The identification of risk conditions for falling is crucial. This study investigated the presence of syncope as a possible cause of falls in older persons admitted to a Sub-Acute Care Unit (SACU) with a diagnosis of accidental fall after initial management in an emergency department and acute hospitalization. *Materials and methods*: A retrospective monocentric study of patients aged ≥65 years, consecutively admitted to a SACU with a diagnosis of fall-related trauma. All patients underwent a complete assessment of the index event and clinical status. Patients were categorized into three groups according to the identified cause of falls: (1) transient loss of consciousness (T-LOC), (2) unexplained fall (UF), and (3) definite accidental fall (AF). *Results*: A total of 100 patients were evaluated. T-LOC was present in 36 patients, UF in 37, and AF in 27. Of the 36 patients with T-LOC, a probable origin was identified in most cases (*n* = 33, 91%), 19 subjects (53%) had orthostatic hypotension, 9 (25%) a cardiac relevant disturbance, 2 (6%) a reproduced vaso-vagal syncope, 2 (6%) severe anemia, and 1 (3%) severe hypothyroidism. The T-LOC group was older and more clinically complex than the other groups. *Conclusion*: In older patients who recently experienced a fall event, the prevalence of syncope is relevant. In frail and clinically complex patients with falls, the identification of the underlying cause is pivotal and can be achieved through prolonged monitoring and a comprehensive assessment of the person.

## 1. Introduction

The overlap between syncope/transient loss of consciousness (T-LOC) and falls is well-established [1,2]. They represent two geriatric syndromes within a continuous spectrum with possible interrelationships. Both are burdened by a negative prognosis in terms of mortality and morbidity in older persons [3]. In addition, they have a substantial impact on the maintenance of independent living [4,5]. Syncope and unexplained falls (UF) occur more frequently with aging, and account for 20–30% of all falls referred to the Emergency Departments (ED) [2,6]. In particular, they are the primary cause of hospitalization in patients with cognitive impairment [4]. More than one-third of older individuals admitted to the hospital for a loss of consciousness experienced one or more falls in the previous year [7,8,9,10]. In older patients with syncope or UF, injuries (including fractures) are common in up to 40% of cases [2]. Overall, the associated costs for medical care are substantial [11,12]. Numerous factors make it difficult to investigate the causes of falls and syncope, especially in the presence of dementia [4], such as incomplete information on the fall, absence of witnesses, together with an ageist attitude to underestimate the relevance of these traumatic events in older persons. The detection of the causes of falls and the identification of individuals with syncope-related falls have important clinical and therapeutic implications that, in turn, may reduce the burden of consequences.

The management of syncope and falls is well standardized in the acute phase [10,13,14,15,16]. In the ED, the discrimination between syncopes and accidental falls and between high- or low-risk syncopes is supported by several diagnostic tools [13,14,15,17,18]. However, even considering that syncope represents a major contributor to a fall [16], when a patient is admitted to the ED for trauma, the attention is often misled by the urgency of the trauma itself, in particular when there is a fracture or a major injury. In addition, during the first evaluation, it is often cumbersome to distinguish between a syncope complicated by a trauma or an accidental fall, as it occurs in older patients, particularly those with cognitive impairment or in the absence of witnesses. It is also worth mentioning that the primary purpose of managing syncope in the ED aims to stratify the risk of adverse outcomes and is primarily focused on the identification of serious underlying causes requiring hospitalization [13,14,15]. In the presence of a trauma requiring hospitalization per se, the etiological ascertaince of the fall would receive a lower priority, being somehow postponed. Then, even during the hospitalization, there is a high risk of understatement of the primary event, i.e., the fall, because of the greater effort required for the treatment of the secondary trauma. This may lead to misdiagnosis without identifying the fall’s source.

It is common for a Sub-Acute Care Unit (SACU) to manage patients with traumatic injuries in the process of consolidation and healing, transferred from an acute hospital ward. As a matter of fact, a SACU represents a hospital unit dedicated to managing older and frail patients after an acute event [19].

Thus, the primary aim of the present study was to evaluate the prevalence of undiagnosed syncope in patients admitted to a SACU with a diagnosis of fall. The secondary aim was to investigate the history and clinical characteristics of these patients with discerning eyes to define the nature of the fall for proper treatment and follow-up.

## 2. Materials and Methods

The study was conducted retrospectively at the SACU of the IRCCS Istituti Clinici Scientifici Maugeri, Milan, over an observation period of 1 year. The study adhered to the principles of the Declaration of Helsinki. According to the IRCCS Maugeri standardized privacy policy, all patients signed a written informed consent for the treatment of their demographic and clinical data for research purposes (Appendix 1). All consecutive patients aged 65 years or older admitted to the SACU with a diagnosis of fall-related trauma and/or fracture were enrolled with data anonymization. Patients were all transferred from an acute medical or surgical ward to our lower intensity care unit for clinical stabilization and treatment.

Upon admission to the SACU, all enrolled patients underwent an evaluation as recommended by the standard of care and guidelines [13,14], which included an in-depth medical interview, physical examination, a 12-lead electrocardiogram (ECG), arterial blood pressure (BP) while supine and during standing (bed-ridden patients were evaluated in supine position only). Supine and active orthostism BP was then routinely checked during the SACU stay. Orthostatic hypotension (OH) was defined as a sustained reduction in systolic BP of at least 20 mmHg or in diastolic BP of 10 mmHg within 3 min of standing. In addition, routine blood tests, Mini Mental State Examination (MMSE, range 0–30, where 30 is normal) [20], Cumulative Illness Rating Scale comorbidity (CIRSc, range 0–14, indicating the number of comorbidities) [21], Cumulative Illness Rating Scale severity (CIRSs, range 1–5, where 1 is absent and 5 is extremely severe) were collected. The witness of the fall, when identified, was also interviewed to collect additional helpful details to better define the circumstances of the event. Specific cardiological tests, such as transthoracic echocardiography; prolonged ECG monitoring; and carotid sinus massage, were also performed during the SACU stay, on the basis of the clinical judgment. Patients were categorized into three groups according to the suspected cause of the fall: (1) T-LOC, (2) unexplained fall (UF), or (3) definite accidental fall (AF). Patients were classified as T-LOC when cerebral hypoperfusion, characterized by a rapid onset, short duration, and complete spontaneous recovery, occurred [4,5,6,7,8,9,10,11,12,13,14] as UF when the fall was not related to extrinsic factors such as poor lighting, unsafe stairways, and irregular floor surfaces or to a precise medical or drug-induced cause [4], as AF when the accidental event was certain and accurately described by the patient or witness.

### Statistical Analysis

One-way analysis of variance (Holm–Sidak test for multiple comparisons), or Kruskal–Wallis one-way analysis of variance on ranks (Dunnet test for multiple comparisons) was applied, when appropriate, to assess the differences between AF and the other groups (i.e., UF and T_LOC). χ^2^ test was applied for categorical variables. The statistical analysis was carried out using the statistical program Sigmaplot (Systat Software, Chicago, IL, USA, version 11.0). A *p* < 0.05 was considered significant.

## 3. Results

Out of a total of 672 patients admitted to the Sub-Acute Care Unit during the study period, 100 (14.9%) were admitted for trauma or fracture (*n* = 46) due to a fall. Of these 100 patients, 70 had been transferred from surgical departments (43 from orthopedics, 27 from general surgery). In none of them, the etiology for the fall event had been established in the acute setting. There were no non-traumatic falls.

Out of the 100 enrolled patients, 36 were categorized as T-LOC, 37 as UF and 27 as AF. Table 1 summarizes the main demographic and clinical characteristics collected upon SACU admission, considering both the total population and the 3 groups defined above, namely T-LOC, UF, and AF.

The distribution of CIRSc and CIRSs in the three groups is detailed in Figure 1. The distribution of MMSE in the three groups is shown in Figure 2.

Of the 36 patients with T-LOC, a relevant and specific pathological condition, possibly contributing to the syncopal episode, was identified in 33 (91%) cases. In eight patients (19%) a major cardiac disease was identified (three had an advanced atrioventricular block, two had paroxysmal atrial fibrillation at a high ventricular rate, one ventricular tachycardia, one severe aortic stenosis, one pulmonary thromboembolism), in two (6%) there was severe anemia, and in one (3%) severe hypothyroidism. Nineteen subjects (53%) had symptomatic OH, of which 10 were drug-induced, and two (6%) had a reproduced vasovagal syncope. One patient died of sudden death. T-LOC patients were older, more clinically complex, had more frequently an abnormal basal ECG, and a history of previous syncope.

## 4. Discussion

The distinction between an accidental fall from a syncope remains an area of great uncertainty, in particular in older persons [1,3,4,5]. Despite the well-codified management of syncope in the ED or during a short observation period [1,13,14,15,16,17,18], the research on falls is limited in quality and quantity [22]. The coexistence of the two conditions is often missed or not adequately handled. When there is a fall event, the focus is often on the trauma, and the investigation of syncope may be frequently neglected. The results of this study, showing the presence of a T-LOC suspected to be syncope in 36% of the studied patients, represent an unpleasant confirmation of this issue. Furthermore, it is reasonable to suppose that the group with UF could include a case-mix of patients with T-LOC and AF. Indeed, the UF group also had more frequent ECG abnormalities and OH than the AF one.

In our population, a fall event probably determined by a syncope occurred to a group of frail patients who, compared to those with accidental falls, were older and had a higher level of physical and cognitive impairment, as shown by comparison of CIRSc/CIRSs and MMSE, respectively. These results are in line with a previous study [4] that considered patients with dementia. All together, the data remark the high difficulty in detecting the cause of a fall in patients with cognitive impairment. In detail, the T-LOC group was characterized by a lower MMSE than the AF group, indicating the presence of cognitive impairment of variable degree.

This study also suggests that the SACU stay might allow identifying the syncope as the likely cause of some fall events otherwise left undiagnosed. It also suggests that the acute setting is often inappropriate for cause analysis and risk stratification in older patients. Conversely, syncope should be identified and stratified by the degree of the risk of severe adverse outcomes over the short- and long-term periods [18]. Thus, a prolonged clinical observation in a low-intensity care unit might overcome this gap, allowing a prolonged ECG telemetry to monitor heart rhythm and multiple evaluations of clinostatic and orthostatic BP. In fact, in the patients of this study, a noteworthy underlying clinical condition was found in 91% of the T-LOC group, corresponding to 31% of the total study sample, mainly cardiogenic disturbances and orthostatic hypotension.

In addition, re-evaluation of heart rate and BP enabled optimization of therapy and drug titration. Indeed, Potential Inappropriate Prescribing (PIP) is often underestimated by physicians [12]. It is extensively documented that most patients present PIP and that 80% of PIP continues even after the hospital discharge, including medications that increase the risk of falls or syncope, specifically, vasodilator drugs and benzodiazepines [12]. Having adequate time to evaluate the appropriateness of the prescribed drugs represents one of the major strengths of the SACU setting, differentiating it from the often-chaotic ED or acute care units. However, in half of the OH patients of this protocol, PIP was not implicated, hinting that OH could be related to some degree of dysautonomia, which is often part of the geriatric syndromes [14,15]. Moreover, patients with dementia deserve additional considerations since they often manifest an atypical pattern of symptoms, a high prevalence of post-event confusion, and retrograde amnesia [4].

### Study Limitations

This is a retrospective study on a limited sample size recruited in a single center. This implies that no causality can be inferred between the diagnoses identified in the T-LOC group and the causes of syncope. However, the results suggest that extended monitoring in patients with a history of falls may help to identify a possible etiology, potentially preventing further episodes and guiding proper follow-up to avoid the recurrence of falls. Further prospective studies on a large elderly population are needed to support our findings.

## 5. Conclusions

In older patients with a traumatic fall, the prevalence of falls likely caused by syncope is relatively high and should always be suspected. The Sub-Acute Care Units may represent an appropriate setting for conducting the needed comprehensive assessment, including a careful evaluation of history, prolonged clinical observation, and appropriate diagnostic testing of the often frail and clinically complex patients with unexplained falls.

## Figures and Tables

**Figure 1 medicina-57-00623-f001:**
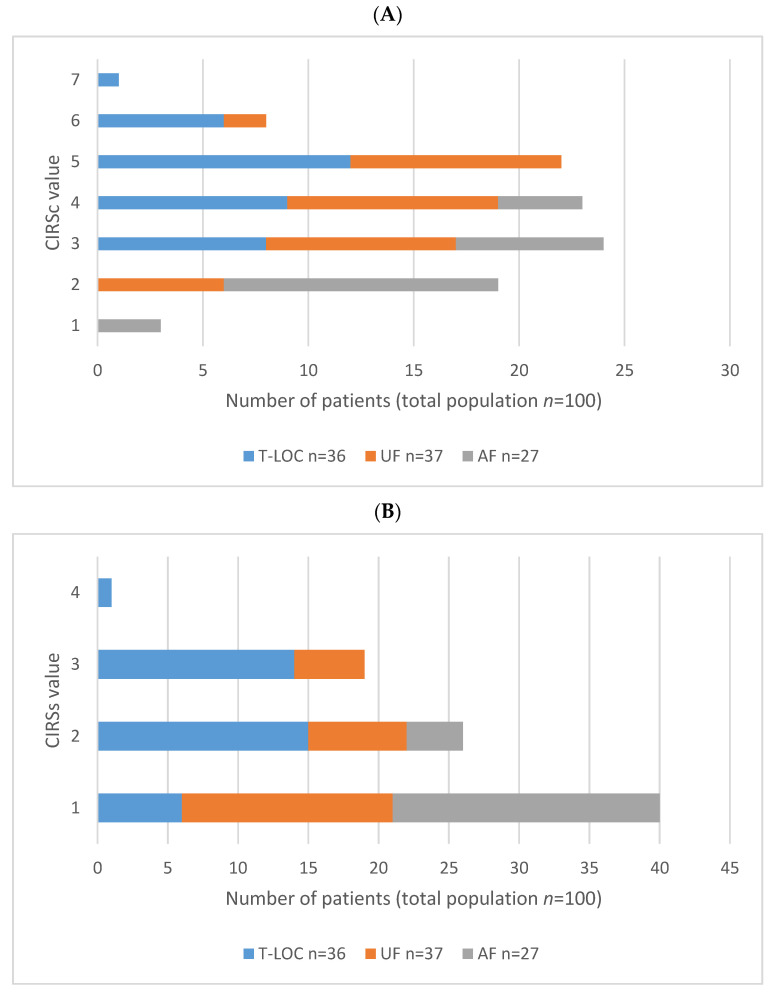
CIRSc (**A**) and CIRSs (**B**) distribution in T-LOC, UF and AF patients. T-LOC: transient loss of consciousness; UF: unexplained fall; AF: accidental fall; CIRSc: Cumulative Illness Rating Scale comorbidity (range 0–14, indicating the number of comorbidities); CIRSs: Cumulative Illness Rating Scale severity (range 1–5, from absent to extremely severe).

**Figure 2 medicina-57-00623-f002:**
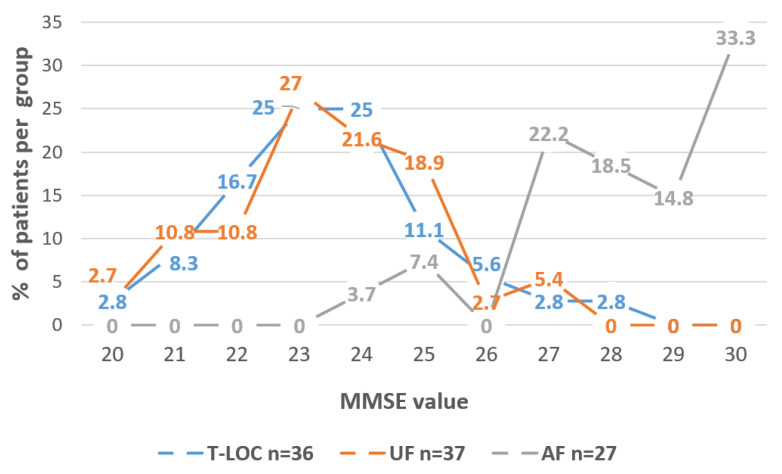
MMSE (range 0–30, where 30 is normal) distribution in T-LOC, UF and AF patients. T-LOC: transient loss of consciousness; UF: unexplained fall; AF: accidental fall.

**Table 1 medicina-57-00623-t001:** Demographic and clinical characteristics of patients admitted to a Sub-Acute Care Unit with a diagnosis of traumatic fall.

	Total	T-LOC	UF	AF
	*n* = 100	*n* = 36	*n* = 37	*n* = 27
Age, yrs	78.4 ± 4.5	83.0 ± 1.4 **	77.0 ± 3.2	74.2 ± 3.1
Sex (women), *n* (%)	61 (61)	24 (67)	20 (54)	17 (62)
Witness, *n* (%)	21 (21)	8 (22)	0 (0) *	13 (48)
Previous syncope, *n* (%)	12 (12)	10 (28) *	2 (5)	0 (0)
Medical Dept, *n* (%)	30 (30)	13 (36) *	12 (32) *	5 (18)
Surgical Dept, *n* (%)	70 (70)	23 (63)	24 (64)	23 (85)
Fracture, *n* (%)	46 (46)	18 (50)	10 (27) *	18 (67)
CIRSc ≥ 4, *n* (%)	58 (58)	28 (78) **	26 (70) **	4 (15)
CIRSs ≥ 2, *n* (%)	20 (20)	15 (42) **	5 (14)	0 (0)
MMSE, score	24.8 ± 2.7	23.5 ± 1.7 **	23.5 ± 1.6 **	28.2 ± 1.7
Heart Rate, bpm	73 ± 11	74 ± 15	74 ± 8	70 ± 8
Abnormal ECG, *n* (%)	55 (55)	27 (75) **	24 (65) **	4 (15)
OH, *n* (%)	69 (69)	29 (81) **	31 (84) **	9 (33)

Data are expressed as mean ±SD or absolute number as appropriate. T-LOC: transient loss of consciousness; UF: unexplained fall; AF: accidental fall; Dept: department; CIRSc: Cumulative Illness Rating Scale comorbidity (range 0–14, indicating the number of comorbidities); CIRSs: Cumulative Illness Rating Scale severity (range 1–5, from absent to extremely severe); MMSE: Mini Mental State Examination (range 0–30, where 30 is normal); bpm: beats per minute; OH: orthostatic hypotension; n: number of patients. * *p* < 0.05 vs. AF; ** *p* < 0.001 vs. AF.

## Data Availability

Data are available upon reasonable request to the corresponding author.

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
