# Peer review of "The Dilemma of Falls in Older Persons: Never Forget to Investigate the Syncope"

_medicina, 2021, doi:10.3390/medicina57060623_

Round 1
Reviewer 1 Report
Thank you for allowing me to review the manuscript entitiled" The dilemma of falls in older persons: never forget to investigate the syncope". This study aims to investigate prevalence of syncope which is masked by an episode of a fall in a Specialized medical unit. Overall I feel this paper has scientific merit but would require major revisions
- The premise and title of the paper appears to be a syncopal episode which the authors have classified as being "masked" by a fall. I am not familiar with this terminology as it has been established that syncope is a cause of falls in older adults. I would suggest the authors revise this terminology and consider something more consistent with terminology used in european guidelines such as syncope as the "major contributory" cause of a fall.
- I think there needs to be more information with regards to how the patients in the study came to be transferred to the SACU. From the paper it states that all consecutive patients were enrolled with a fall related truama or fracture. This would imply to me that all of the patients studied in this paper had experienced their first injury or fracture related to a fall? Do all patients who have a fal related trauma or fracture get transferred to this unit (i.e 100% would be transferred). Could there be other patients who remain on a truma unit or get transferred directly to a higher level of care? This is important as the stated aim was to establish the prevalence of syncope. Were there any patients admitted to the unit who suffered a fall but did not suffer an injury associated with the fall? The results section does not clarify this for me and I am left confused as to the basic questions of how many patients were admitted to the SACU with a fall, How many of these had T-LOC, How many of these of these suffered a fracture or injury? There appears to be multiple different numbers of patients who were studied and I am finding it difficult to find the exact prevalence as a result
- I believe you mean arterial blood pressure (line 79 page 2).
- No definition provided as to how you defined a fall or the sub-groups that are essential to the analysis of the paper.
- It does not appear that the patients underwent a tilt table test or other provoking test?
- It appears from the results that you had included a variety of causes of T-LOC including pulmonary embolism, and death. This is again a very large variety of things that were included as a possible cause of T-LOC and it may convey a misunderstanding as to what consititues T-LOC. I would urge the authors to revise the ESC 2018 defintion of T-LOC as they would find that things such as sudden death are not part of this defintion.
- How did you decide who to perform a carotid sinus massage on? This is important for the prevalence of syncope.
- I would suggest that the authors reduce the size of the introduction and focus on the background of the reason why they would study this - i.e why does it matter if syncope is the major conritbutory cause of a fall.
Reviewer 2 Report
The submitted manuscript by Perego et al. investigated the prevalence of syncope as a potential cause of falls in elderly patients admitted to the subacute care unit for a traumatic injury. Furthermore, the authors examined between-group differences of syncope-related falls compared to unexplained and accidental falls. The results underline a high prevalence of syncope-related falls and indicate that older, more frail patients with higher levels of physical and cognitive impairment are associated with syncope-related falls compared to falls without syncope. Thus, subacute care units are a good way to examine potential underlying conditions associated with syncope.
Broad comments:
While the introduction mentions important studies and provides a general overview of syncope-related falls and the health consequences, it is sometimes hard for the reader to follow the storyline and to fully understand the relevance of this study. The authors should therefore aim to improve the flow and readability of this part of the manuscript, providing reasons for the necessity of this study. Ideally, the reader should know the study aim without having read it up.
Considering available literature (Bhangu et al. 2019; Ungar et al. 2016), the prevalence of syncope-related falls in elderly patients alone does not seem to constitute sufficient novelty to be the primary study outcome. However, between-group differences as investigated by the authors would add to the existing literature. Thus, the study aim should be adapted accordingly. Lastly, even though, the between-groups comparison is a major part of the manuscript, this is not mentioned in the introduction. A rationale for these analyses would strengthen the manuscript and highlight the importance of this research.
The methods used to answer the research question of this study seem appropriate.
Regarding the discussion of the results, a comparison of the findings with available literature (e.g. Ungar et al. 2016) would enhance the quality of the manuscript and help the reader to understand the results.
Finally, the authors should be aware that it is not possible to imply causality from the study design (retrospective) used. Therefore, the conclusions made about e.g. orthostatic hypotension or cardiac origin need revision.
Specific comments:
Lines 48 & 49: "Indeed, drug prescriptions [...]." This seems out of context and might be more suitable for the discussion.
Lines 56 & 57: The passage is not intuitive. A statement is made but the consequence of this decline in physical function is not discussed. Thus, the reader themself has to come up with why this is relevant.
Materials and methods: Please provide information about whether the study received ethical approval.
Lines 83-84: Please provide a short explanation of the scales used in text tables and figures, i.e. higher score = worse health. This will facilitate the interpretation of the findings for the reader.
Line 84: Please explain how the information collected during the interviews with the witnesses was used.
Lines 88 & 89: A definition of each group (T-LOC, AF, UF) is necessary to make the study reproducible and clear to the reader. As an example please see Ungar et al. (2016).
Lines 98-102: Table 1 is not referred to in the text. Please include a reference.
Line 103: Please explain the abbreviation "SACU" or use "subacute care unit" instead. This will ensure that the reader understands the tables and figures without having to read the manuscript.
Table 1: "Physical therapy" - I do not understand how this information is relevant to the manuscript. It was not explained in the text. Please include an explanation or remove it.
Figure 2: Numbers at MMSE-value 20 are not properly visible and this type of graph might be misleading. Consider changing it to e.g. a grouped bar chart. If this should take up too much space, at least adapt the placement of the values in the graph.
References: Please use a consistent reference style according to the journal guidelines. E.g. publication year in lines 253 & 255 is not bold or no full-stop after each word in the journal name.
Author Response
please, see the attachment. Thank you very much.

Round 2
Reviewer 2 Report
All comments were discussed by the authors and targeted appropriately. I feel the quality of the manuscript has improved substantially. Yet, I do suggest conducting a thorough spell-check. Please find some non-exhaustive suggestions below.
Abstract:
Falls [...]. (L. 10)
This study investigated [...]. (L. 12)
A retrospective [...]. (L. 14)
In older [...]. (L. 23)
Full-text:
[...] falls referred to [...]. (L. 35)
[...] when a patient is referred to [...]. (L. 50)
range 0-30, [...]. (L. 89)
Sub-Acute [...]. (L. 121)
Figure 2. Consider matching the axes titles in terms of style with figure 1 (i.e. not all caps). Furthermore, if I understood correctly, the y-axis title should state "% of patients per group".
[...] investigation of syncope may be [...]. (L. 163)
L. 191: Please provide references for this statement.
L. 208-209: Would the following structure make more sense?
In older patients with a traumatic fall, the prevalence of falls likely caused by syncope is relatively high [...].
Author Response
We thank the reviewr #2 once again.
We changed the text as suggested, the figure 2 was also modified accordingly.
The manuscript was rechecked and some typos were fixed.
All new changes are in red.